# Supplementation with Octacosanol Affects the Level of PCSK9 and Restore Its Physiologic Relation with LDL-C in Patients on Chronic Statin Therapy

**DOI:** 10.3390/nu13030903

**Published:** 2021-03-10

**Authors:** Milica Zrnic Ciric, Miodrag Ostojic, Ivana Baralic, Jelena Kotur-Stevuljevic, Brizita I. Djordjevic, Stana Markovic, Stefan Zivkovic, Ivan Stankovic

**Affiliations:** 1Department of Bromatology, Faculty of Pharmacy, University of Belgrade, Vojvode Stepe 450, 11221 Belgrade, Serbia; brizitadjordjevic@gmail.com (B.I.D.); istank@pharmacy.bg.ac.rs (I.S.); 2Faculty of Medicine, University of Belgrade, 6 Dr Subotica Street, 11000 Belgrade, Serbia; mostojic2003@yahoo.com; 3Institute for Cardiovascular Diseases “Dedinje”, Milana Tepica 1, 11000 Belgrade, Serbia; 4Department of Cardiology, University Clinical Centre of the Republic of Srpska, 78000 Banja Luka, Bosnia and Herzegovina; 5Medical Faculty, University of Banja Luka, 78000 Banja Luka, Bosnia and Herzegovina; 6Department of Pharmacy, Zvezdara University Medical Center, Dimitrija Tucovića 161, 11000 Belgrade, Serbia; ivanabaralic111@gmail.com (I.B.); stana.markovic02@gmail.com (S.M.); zivkovic.stefan@gmail.com (S.Z.); 7Department of Medical Biochemistry, Faculty of Pharmacy, University of Belgrade, Vojvode Stepe 450, 11221 Belgrade, Serbia; jkotur@pharmacy.bg.ac.rs

**Keywords:** supplementation, Octacosanol, LDL-C, PCSK9, statins

## Abstract

Dietary supplementation with sugar cane derivates may modulate low-density lipoprotein cholesterol (LDL-C) and proprotein convertase subtilisin/kexin type 9 (PCSK9) levels. The purpose of this study was to determine if dietary supplement (DS), containing Octacosanol (20 mg) and vitamin K2 (45 µg), could restore the disrupted physiologic relation between LDL-C and serum PCSK9. Double-blind, randomized, placebo-controlled, single-center study including 87 patients on chronic atorvastatin therapy was conducted. Eighty-seven patients were randomized to receive DS (*n* = 42) or placebo (*n* = 45), and followed for 13 weeks. Serum PCSK9 levels, lipid parameters and their relationship were the main efficacy endpoints. The absolute levels of PCSK9 and LDL-C were not significantly different from baseline to 13 weeks. However, physiologic correlation between % change of PCSK9 and % change of LDL-C levels was normalized only in the group of patients treated with DS (*r* = 0.409, *p* = 0.012). This study shows that DS can restore statin disrupted physiologic positive correlation between PCSK9 and LDL-C. Elevated PCSK9 level is an independent risk factor so controlling its rise by statins may be important in prevention of cardiovascular events.

## 1. Introduction

Data on the lipid-lowering effects of policosanol (alone or in combination with statins) is contradictory [1], but high-quality studies have disputed any favorable effects on serum low-density lipoprotein cholesterol (LDL-C) levels [2,3,4,5,6]. It is not clear if genetic differences are responsible for such a diversity of results, but the use of policosanol has been discouraged in the current European guidelines for the management of dyslipidemias [7]. Nonetheless, it is widely available as a dietary supplement (alone or in combination with different nutraceuticals) all over the world. 

Proprotein convertase subtilisin/kexin type 9 (PCSK9) and PCSK9 inhibitors have gained a lot of attention in the last decade. PCSK9 is a protease that promotes lysosomal degradation of LDL receptor (LDLr), thus regulating the amount of LDL-C in circulation [8]. This mechanism of action of circulating PCSK9 has initiated the development of a novel generation of drugs (PCSK9-inhibitors), which effectively block the PCSK9-LDLr interaction and lower LDL-C levels up to 70% [9]. In addition, investigation of the statins-PCSK9 interactions revealed an increase in serum PCSK9 levels as a result of statin therapy, which could explain reduced full lipid-lowering effects of statins [10,11]. Interestingly, it has been suggested that policosanol may prevent the rise of PCSK9 levels in patients who start statin therapy, as well as decrease them in healthy volunteers [12]. Also, several nutraceuticals may decrease PCSK9 levels in animals and humans [13,14]. Nutraceuticals, such as berberin and curcumin, may inhibit PCSK9 expression through SREBP (sterol-responsive element binding protein)-independent pathways [15,16,17]. In contrast, statins increase the expression of PCSK9 through the activation of SREBP-2, which may attenuate their beneficial effects [18,19]. However, it is not clear if elevated PCSK9 levels (irrespective of LDL-C values) represent an independent risk factor for future cardiovascular events [20,21] or not [22,23]. It would be the issue even with the new very recently approved drug Inclisiran (slow interfering RNA, siRNA) which inhibits PCSK9 synthesis because it has been administered on top of background statin therapy [24].

Therefore, we investigated in a prospective, single-center, randomized, double-blind, placebo-controlled trial if a dietary supplement, which contains Octacosanol (20 mg) and vitamin K2-MK7 (45 µg) may influence the level of PCSK9 and its relationship with LDL-C in patients on chronic statin therapy.

## 2. Materials and Methods

### 2.1. Study Population

One-hundred and seventy-seven outpatients from the Department of Cardiology, University Clinical Centre Zvezdara, Belgrade, Serbia were initially screened for eligibility. Finally, 87 patients were examined and subjected to blood analyses to determine lipid profile parameters and complete biochemistry. 

The inclusion criteria were: Both sexes, between the ages of 40–80 years, confirmed history/diagnosis of hypercholesterolemia or mixed dyslipidaemia, body mass index (BMI) between 18 and 35 kg/m^2^, and the use of atorvastatin (20 mg/day) for a minimum of 4 months prior to study entry. The exclusion criteria included serum triglycerides (TG) levels above 5.6 mmol/L, acute coronary syndrome within the previous month, serious heart failure, cerebral vascular disease, history of severe infections, known hypersensitivity to any of the ingredients of the formulation, current use of agents with a potential to interact with Octacosanol (e.g., high-dose aspirin), recent or chronic use of oral anticoagulant drugs and anticipated compliance problems. Significant pre-existing diseases including cancer, liver and/or renal insufficiency, psychiatric disorders, systematic inflammatory or autoimmune disease were also the exclusion criteria. In addition, patients with serum alanine aminotransferase (ALT) and/or aspartate aminotransferase (AST) levels 3 times above the upper limit of the normal laboratory range, and/or creatine phosphokinase (CK) 5 times above the upper limit of normal were not included [25]. 

### 2.2. Study Design and Protocol

From November 2015 to June 2016 in randomized, double-blind, placebo-controlled trial, subjects were assigned to receive one capsule daily of dietary supplement or placebo. The investigated supplement and placebo were produced and provided by AbelaPharm (Belgrade, Serbia). Dietary supplement contains Octacosanol (20 mg) and vitamin K2-MK7 (45 µg) (Arteroprotect^®^; Gnosis S.p.A, Italy). The placebo capsules were identical in appearance (color, shape, texture) and taste to the supplement but consisted of magnesium stearate, microcrystalline cellulose and colloidal silicon dioxide. The applied dose of 20 mg of Octacosanol corresponds to approximately 30 mg of Policosanol. There is no report on the use of higher Octacosanol doses alone, but previous study used 80 mg of Policosanol which would correspond to 50 mg of Octacosanol [3]. 

Patients were allocated by a computer-generated random sequence of numbers run by the investigator with no involvement in the study. Only the safety monitor (DK), who was otherwise uninvolved in the study, could unblind the data.

Subjects were instructed to avoid the use of vitamin K and dietary supplements that can interfere with serum LDL-C (specifically sterol and stanol products, psyllium-based fiber supplements, or red yeast rice) or TG levels (fish oil, omega-3 fatty acids), 1 month prior to the start of the study and during the study.

The recommended low-fat diet and other medications were monitored by a 3-day food/medication record kept at the baseline and at the end of the study. The patients were examined and subjected to blood analysis at the baseline, and after 8 and 13 weeks. During these visits, as well as on the 5th and 10th week, patients were interviewed about adverse effects. Instructions for use were reconfirmed each week by the same investigator. Acceptable patient compliance was defined as an overall dietary supplement /placebo intake of at least 85% of scheduled capsules, which was verified by the number of returned capsules at the end of 8th and 13th week. 

The study was approved by the Zvezdara University Medical Center Office for Human Research Protections (ethics code number 27102015; date 27 October 2015). The study was conducted according to the Declaration of Helsinki. All patients gave their written consent after being informed about the purpose, demands, and possible risks associated with the study. The trial was registered at Australian New Zealand Clinical Trials Registry as ACTRN12619000102178; http://www.anzctr.org.au (accessed on 19 January 2021).

### 2.3. Outcomes

Serum PCSK9 concentrations were measured in duplicate using a commercial, high-sensitivity, quantitative sandwich enzyme immunoassay Quantikine^®^ (R&D System, Minneapolis, MN, USA). This assay recognizes free and LDL receptor bound PCSK9. The intra- and inter-assay coefficients of variation vary from 4.1% to 6.5%, and the range for serum values in a random sample of healthy individuals was 177–460 ng/mL, with a mean ± S.D. value of 313 ± 71 ng/mL. The mean threshold for detection was 0.096 ng/mL. 

Additionally, we evaluated the concentrations of lipid parameters including total cholesterol (TC), high-density lipoprotein cholesterol (HDL-C), LDL-C, TG, apolipoprotein A1 (ApoA1), and apolipoprotein B100 (ApoB100), as well as AST, ALT, CK, C-reactive protein (CRP), and glucose. All biochemical parameters were measured using an automated biochemistry analyzer (Hitachi 7150, Tokyo, Japan) employing commercial kits (Boehringer, Mannheim, Germany). 

Blood sample collection was performed between 9:00 and 10:00 a.m., after at least 8 h of overnight fasting.

### 2.4. Safety Evaluation

Physical and clinical examinations, vital signs (pulse rate, diastolic and systolic blood pressure) and laboratory indicators (ALT, AST, fasting glucose, and CK) were evaluated for safety assessment. At each visit any unusual adverse effects was reported by using appropriate record forms.

### 2.5. Statistical Analysis

For the sample size estimation, we used a previously published study [12] which demonstrated a 22% lower increase in serum PCSK9 levels when atorvastatin was given with policosanol vs. atorvastatin and placebo. For a level of significance of 5% and power of 80%, a sample size of 76 patients would be sufficient to detect this level of effect. Allowing for an estimated dropout rate of 10%, recruitment of 84 patients was needed. The required sample size was estimated using the Power Analysis and Sample Size software version 11.0.10 (PASS, NCSS statistical software, LLC, Kaysville, UT, USA).

All statistical analyses were performed with SPSS version 20.0 software (Chicago, IL, USA). All data were assessed for normality. The results are expressed in terms of mean values and standard error of mean (SEM) for normally distributed variables, geometric means and 95% confidence intervals (CI) for variables which were normally distributed after logarithmic transformation, median, and interquartile range for variables with a skewed distribution even after logarithmic transformation, and as number (percentage) for categorical variables. Percentage change was calculated by taking the difference between the value at 8 or 13 weeks and the basal value, divided by basal value and multiplied by 100. To compare subjects’ baseline characteristics between the treatment groups, we used an independent-sample t-test for normally distributed parameters and Mann–Whitney U-test for parameters that were not normally distributed. Chi-squared-test was used to analyze the differences in categorical data. ANOVA with repeated measurements was used to describe the interaction of time and supplementation, effects of investigational supplement and changes over time (followed by post-hoc comparisons with Bonferroni corrections). Pearson correlation coefficients were calculated in order to describe the correlation between the examined parameters. A *p* value < 0.05 was regarded as statistically significant.

## 3. Results

### 3.1. Patient Characteristics

We consecutively screened 177 patients and then enrolled 87 patients, according to the inclusion/exclusion criteria. Out of 87 patients, 45 (51.7%) and 42 (48.3%) were randomly assigned to receive placebo or dietary supplement for 13 weeks, respectively. Six patients dropped out before the endpoint of the study, 5 patients from the dietary supplement group and 1 patient from the placebo group because of personal reasons not related to the study and due to gastric distress (Figure 1). 

Baseline characteristics of the patients are shown in Table 1. The mean age of the patients was 62.6 ± 0.8 years, with males representing 63.2% of the participants. The mean BMI was 27.6 ± 0.4. More than half of the subjects (52.9%) had a family history of coronary heart disease. History of dyslipidemia was present in all patients. Overall, patients with hypertension, diabetes and current smokers represented 96.6%, 26.4%, and 24.1% of the study population, respectively. Previous myocardial infarctions and myocardial revascularization (predominantly by stents) were present in 60% and 62% of patients, respectively. There were no significant differences in demographic and anthropometric parameters, smoking status or any of the clinical variables between the groups. There was also no significant difference in most biochemical parameters between the two groups, except for TG levels (which were higher in the placebo group, *p* < 0.05). Concomitant medications were also equally distributed among the groups. Adherence was 97%, as defined above.

### 3.2. Effects on PCSK9 and Lipids Parameters

Table 2 shows serum PSCK9 levels and lipid parameters at baseline, after 8 and 13 weeks. PCSK9 levels remained constant during the study period in both groups. The TC and ApoA1 levels did not significantly change over the study period in both groups. There was a statistically significant decrease in LDL-C levels in the placebo group after 8 weeks compared with baseline (interaction effect of supplementation and time, *p* = 0.023), but the effect disappeared after 13 weeks. Baseline serum levels of TG were significantly higher in the placebo group than in the dietary supplement group (*p* = 0.012). The serum levels of TG significantly increased after 8 weeks compared with baseline in the dietary supplement group (main effect of time, *p* = 0.007), but the effect disappeared after 13 weeks. No changes from baseline HDL-C values were observed in patients treated with dietary supplement over the 13-week period, whereas an increase in HDL-C values was observed in the placebo group (*p* < 0.001, Bonferroni post-hoc test). The serum level of ApoB100 increased significantly (but clinically not relevant) after 8 weeks compared with baseline in the dietary supplement group, but the effect disappeared after 13 weeks (main effect of time, *p* = 0.025). 

A Pearson correlation analysis did not show any significant correlation between PCSK9 levels and age, BMI, TC, LDL-C, HDL-C, ApoA1, ApoB100, and TG either at baseline or after 13 weeks of intervention regardless of whether patients were using atorvastatin alone or in combination with dietary supplement.

We observed a significant positive correlation between the percentage change in PCSK9 levels and those in LDL-C levels as soon as after 8 weeks (*r* = 0.383, *p* = 0.019), and it remained at week 13, i.e., from baseline to endpoint (*r* = 0.409, *p* = 0.012) (Figure 2A,C) in the dietary supplement group. Such a correlation was not detected in the placebo group (week 8—*r* = 0.078, *p* = 0.615; week 13—*r* = −0.103, *p* = 0.508) (Figure 2B,D). Therefore, supplemented subjects that had the greatest decrease in PCSK9 levels tended to have the largest decrease in serum LDL-C levels. 

Significant negative correlation between the percentage change in PCSK9 levels and those in HDL-C levels was found after 8 weeks (*r* = −0.369, *p* = 0.025), but this correlation was lost at week 13, i.e., from baseline to endpoint (*r* = −0.276, *p* = 0.098) (Appendix A) in patients with combination of statin and the dietary supplement. Such a correlation was not detected in the placebo group (week 8—*r* = −0.105, *p* = 0.498; week 13—*r* = −0.063, *p* = 0.686) (Appendix A). Therefore, supplemented subjects that had the greatest decrease in PCSK9 levels tended to have the largest increase in serum HDL-C levels after 8 weeks but this correlation disappeared after 13 weeks

A total of 13/37 (35%) of dietary supplement—treated and 22/44 (50%) of placebo-treated patients had favorable changes in lipid profile defined as decrease in LDL-C and increase in HDL-C after 13 weeks. Figure 3A illustrates the changes in serum PCSK9 levels of these participants from baseline to endpoint. No significant changes in the PCSK9 levels were observed within the two subgroups by the end of treatment period. Nevertheless, the two treatments performed differently. The PCSK9 levels at baseline were comparable in these two subgroups: 236.26 ± 31.48 ng/mL in patients taking dietary supplement and 241.11 ± 25.34 ng/mL in patients receiving placebo (*p* > 0.05). However, at the end of the 13th week, the PCSK9 values tended to be significantly lower in patients receiving dietary supplement (280.33 ± 33.96 in the placebo group vs. 191.47 ± 25.85 in the supplemented group, *p* = 0.077). 

PCSK9 levels during the study period in subjects with non-favorable lipid profile changes are shown in Figure 3B. No significant changes in PCSK9 levels were observed in the dietary supplement group nor placebo group.

### 3.3. Safety

AST levels significantly decreased in both groups, with a more pronounced decrease in the dietary supplement group (main effect of time, *p* < 0.001). In addition, supplementation with dietary supplement significantly decreased ALT levels by the end of the study period (interaction effect of supplementation and time, *p* < 0.05). CK values did not significantly change over the study period in both groups (Appendix A).

## 4. Discussion

The present study was conceived and undertaken to investigate the potential additive or synergistic effects of dietary supplement on serum PCSK9 levels and lipid profile parameters when used with background long term atorvastatin therapy at the time when there was no single report indicating that elevated PCSK9 might be independent health prognostic factor regardless of LDL-C level. The two groups were well matched in terms of all variables, except TG levels. Almost 80% of the patients had a history of ischemic heart disease and were eligible for secondary prevention of cardiovascular events. The study population (*n* = 87) was large enough to address the objectives of the study. Our results showed no adverse effects in the study population regarding ALT and AST levels (in fact, a decrease of these enzymes was observed) as well as on CK levels. 

The main findings of our study, outlined according to originality and importance are as follows. 

First, the most striking and interesting result is the presence of a positive correlation between percentage changes in PCSK9 and those in LDL-C levels as early as after 8 weeks, remaining at 13 weeks in the supplemented group (week 8—*r* = 0.383, *p* = 0.019; week 13—*r* = 0.409, *p* = 0.012). Therefore, supplemented subjects that had the greatest decrease in PCSK9 levels tended to have the largest decrease in serum LDL-C, which is in contrast to all statin studies [10,11,26]. More specifically, several previous studies have found that the greatest increase of PCSK9 levels is seen in patients with the greatest response to statin therapy [11,27]. There is a significant negative correlation between the percentage change in PCSK9 levels and those in HDL-C levels was found after 8 weeks (*r* = −0.369, *p* = 0.025), but this correlation was lost at week 13 in patients with combination of statin and the dietary supplement. Actually, dietary supplement reestablished the disrupted physiologic relationship between LDL-C and PCSK9 levels, induced by chronic statin use. To the best of our knowledge, this is the first study which has demonstrated described effect on humans. 

Second, in both patient groups, the absolute levels of PCSK9 did not change significantly over the study period. In the study by Guo et al., 36 patients, who had no history of statin use within the previous month, were assigned to receive either atorvastatin (*n* = 17; 5 patients lost of follow up) or atorvastatin plus policosanol (*n* = 19; 5 patients lost of follow up). The study showed that during 8 weeks of atorvastatin monotherapy serum PCSK9 levels significantly increased (by 39.4%) in comparison to pretreatment values (*p* = 0.002). However, in the group which received policosanol with atorvastatin PCSK9 levels increased by 17.4%, without statistical significance when compared to basal values (*p* = 0.184) [12]. This finding indicates that policosanol might attenuate statin-induced increases in serum PCSK9 levels in statin-naive patients. In healthy volunteers (*n* = 7), treated with policosanol for 12 weeks, a decrease of PCSK9 levels was observed, but did not reach statistical significance (*p* = 0.069) [12].

In a post-hoc explanatory analysis, when the patients were classified as having favorable changes of lipid parameters (LDL-C decrease and HDL-C increase) [28], a statistical trend towards PCSK9 decrease in the supplemented group as well as PCSK9 increase in the placebo group after 13 weeks of intervention, has been found. Difference in PCSK9 levels, from baseline to 8 and 13 weeks, between the supplementation and placebo group showed a strong trend of PCSK9 decrease only in patients with favorable lipids changes in the supplemented group (*p* = 0.077). It might be that these patients had physiological regulatory changes of PCSK9 and LDL-C, which restored the positive correlation between them. Described trend was absent in 50% of patients in the placebo group who had favorable changes in the lipid profile (Figure 3A). 

Certain genetic differences could explain different response in patients supplemented with Octacosanol/vitamin K2 regarding PSCK9 and LDL-C levels. Concomitant fall in PCSK9 and LDL-C after the supplementation could be determined by PCSK9 polymorphism which enabled octasosanol/vitamin K2 to restore positive correlation between them. In the study investigating the effect of multicomponent dietary supplement (policosanol, fermented rice with red yeast, berberine, coenzyme Q10, folic acid, and astaxanthin) in people with moderate hypercholesterolemia, a large interindividual variability in the LDL-C response to supplementation was observed [29]. It was shown that LDLR and PCSK9 polymorphisms were associated with response to supplementation.

There was a statistically significant decrease in LDL-C levels in the placebo group after 8 weeks (but not after 13 weeks) compared with baseline. These reductions were around 10% and therefore of limited clinical relevance. Of note is the fact that in other studies, where patients were treated with statins and/or PCSK9 inhibitors, some fluctuations of LDL-C levels over time were observed [4,30]. It is not clear if these fluctuations reflected some dietary changes or were due to some other factors. 

Third, dietary supplement, as expected, did not offer additional lipoprotein benefits when added to background statin therapy (Table 2), similarly to previous studies which examined the combined use of policosanol and statins [2,4]. It is possible that we did not detect a significant effect of dietary supplement on LDL-C in our patients, because they were already decreased by statin use. This is supported by the evidence which suggests that baseline LDL-C levels could determine overall LDL-C reduction [31,32]. 

Although dietary supplement in our study did not show any lipid lowering benefits, its combination with atorvastatin may be useful in an attempt to avoid potential adverse effects that are associated with statin-induced increases of PCSK9 level. The hypothesis that the prognosis of patients will be better with lower LDL-C levels, and while keeping PCSK9 levels under control (which was found in our study for some patients receiving dietary supplement), remains to be confirmed in larger studies [20,21].

Interestingly, in one study of the general population PCSK9 levels demonstrated a week correlation with LDL-C (r^2^ = 0.06, explaining less than 6% of the variation in LDL-C) [33]. However, despite the low aforementioned coefficient of determination (0.06) the PCSK9 inhibitors have been proven to be very effective in decreasing LDL-C (up to 60% from basal values). Having in mind such exemplary discordance, one should view the significance of our results where PCSK9 levels explained LDL-C by 17% (*r* = 0.409) in patients receiving dietary supplement. One possible explanation may be that that quality, not the quantity of PCSK9, is of higher importance. 

The strengths of the present study were as follows: The study design (prospective, randomized, double-blind, placebo-controlled), a high level of compliance and only six participant withdrawals. PCSK9 levels were measured using a validated methodology and our values are in line with a recently reported study [21]. LDL-C levels were measured, not calculated as in many studies. The investigated supplement possessed a good safety profile and tolerability. 

Our study had certain limitations. It was a single-center study and we enrolled only patients who had been taking atorvastatin for 4 months, without a group of statin-naive patients as we took the findings by Guo et al. as granted [12]. It is possible that the simultaneous start of policosanol and statin would have been more effective at regulating PCSK9 levels than adding policosanol to chronic statin therapy. Also, just one dose of Octacosanol was used. Additionally, we did not measure either the level of atorvastatin, or the level of Octacosanol/Vitamin K2 in the serum, but the lipid profiles of our patients in the time course were suggestive of excellent compliance with statin use. The fact that disrupted association between LDL-C and PCSK9 levels was reestablished only in the supplemented group in both time points, indicates that patients were compliant with the therapy. We used this particular supplement because it was available as an over-the-counter product, therefore production and/or regulatory restrictions were not present. We could not exactly distinguish the effect of Octacosanol vs. vitamin K2, but all the available data indicate that low-dose vitamin K2 (45 µg) does not have any lipid regulating effects. However, vitamin K2 has a potential health benefit for cardiovascular disease prevention, related to protection against vascular calcification [34,35]. 

A larger sample and a longer period of supplementation could provide more specific results regarding Octacosanol effects as well as its underlying mechanism in PSCK9 regulation. It could also enable a generalization of the study. Since, this was an investigator-initiated study with zero funding, it was not possible to include a larger number of patients.

Last but not least, the clinical endpoints of the reduction of PCSK9 levels induced by dietary supplement will need to be documented. This is of particular importance, considering that PCSK9 inhibitors are highly effective at decreasing LDL-C, but have a very limited effect in decreasing clinical events, especially mortality. 

## 5. Conclusions

This is the first study to show that dietary supplement can restore the physiologic positive correlation between PCSK9 and LDL-C levels, disrupted by statin therapy. Treatment with statins and/or PCSK9 inhibitors provokes an increase of PCSK9 and a negative correlation between PCSK9 and LDL-C levels. Our finding could be clinically relevant, considering that spontaneous and/or drug-induced elevations of PCSK9 levels may be an independent health risk factor.

## Figures and Tables

**Figure 1 nutrients-13-00903-f001:**
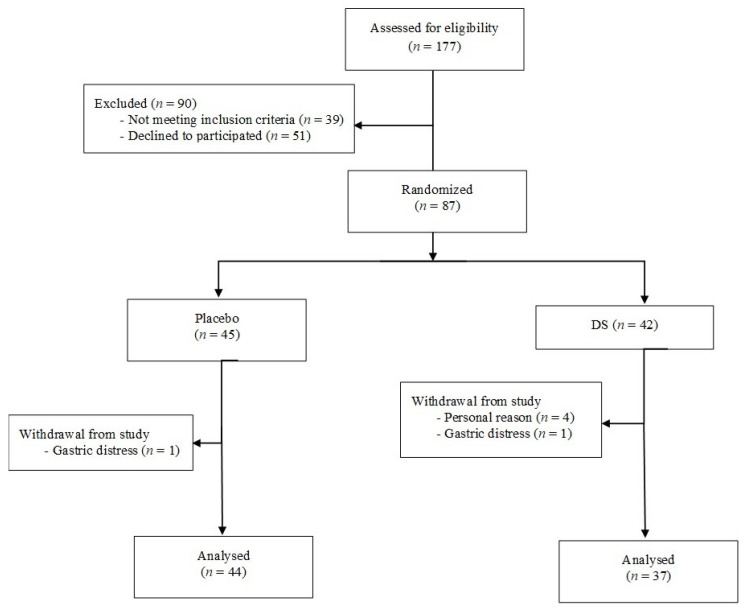
Flow diagram of participants through the study.

**Figure 2 nutrients-13-00903-f002:**
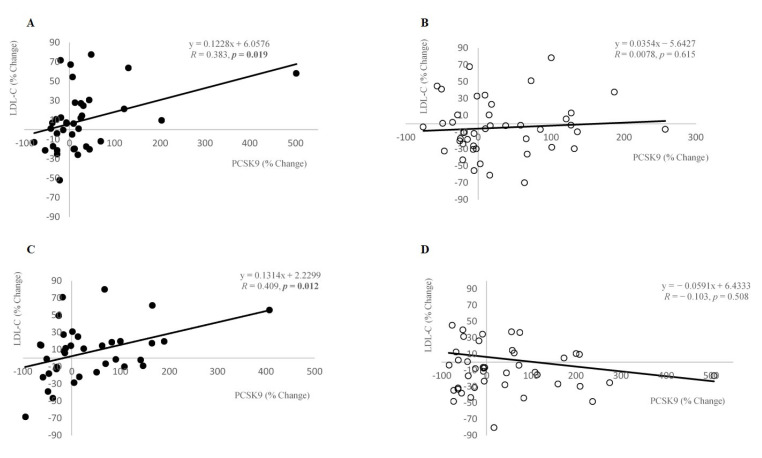
Comparison of atorvastatin-induced percentage changes in serum PCSK9 levels with percentage changes in serum LDL-C levels at 8 weeks (**A**,**B**) and 13 weeks (**C**,**D**). White circles (○) indicate the placebo and black circles (●) indicate supplement. Abbreviations: LDL-C, low density lipoprotein cholesterol; PCSK9, proprotein convertase subtilisin kexin type 9.

**Figure 3 nutrients-13-00903-f003:**
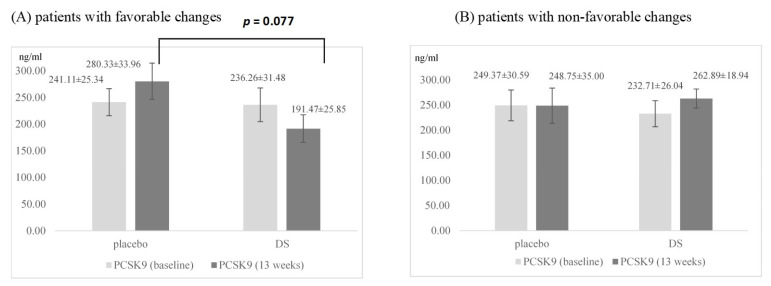
Serum PCSK9 levels at baseline and after 13 weeks of supplementation in patients (**A**) with favorable changes of LDL-C and HDL-C and (**B**) with non-favorable changes of LDL-C and HDL-C. Columns represent mean group values ± SEM. Abbreviations: HDL-C, high density lipoprotein cholesterol; LDL-C, low density lipoprotein cholesterol; PCSK9, proprotein convertase subtilisin kexin type 9.

**Table 1 nutrients-13-00903-t001:** Baseline characteristics of the study population.

Variables	All Patients (*n* = 87)	Placebo (*n* = 45)	Dietary Supplement (*n* = 42)	*p* Value
Age (years)	62.6 ± 0.8	61.5 ± 1.2	63.8 ± 1.1	0.165
Sex				0.806
male	55 (63.2)	29 (64.4)	26 (61.9)	
female	32 (36.8)	16 (35.6)	16 (38.1)	
Body weight (kg)	80.68 ± 1.35	81.05 ± 1.85	80.27 ± 2.00	0.774
Height (cm)	1.71 ± 0.01	1.71 ± 0.01	1.71 ± 0.01	0.773
BMI (kg/m^2^)	27.6 ± 0.4	27.8 ± 0.5	27.4 ± 0.6	0.604
WC (cm)	97.65 ± 1.08	98.44 ± 1.48	96.79 ± 1.59	0.446
WHR	0.92 ± 0.01	0.93 ± 0.01	0.90 ± 0.01	0.090
Family history of				
coronary heart disease	46 (52.9)	26 (57.8)	20 (47.6)	0.343
Smoker	21 (24.1)	11 (24.4)	10 (23.8)	0.945
Diabetes mellitus	23 (26.4)	12 (26.7)	11 (26.2)	0.960
History of hypertension	84 (96.6)	43 (95.6)	41 (97.6)	0.598
Previous myocardial				
infarction	52 (59.8)	29 (64.4)	23 (54.8)	0.357
Coronarography	63 (72.4)	33 (73.3)	30 (71.4)	0.843
Stent	54 (62.1)	28 (62.2)	26 (61.9)	0.976
Bypass	4 (4.6)	2 (4.4)	2 (4.8)	0.944
Primary prevention	20 (23.0)	9 (20.0)	11 (26.2)	0.493
CRP (mg/L) †	1.44 (0.90–2.45)	1.59 (0.88–2.45)	1.34 (0.96–2.37)	0.865
Glucose (mmol/L) ‡	5.64 (5.14–6.19)	5.62 (4.94–6.28)	5.72 (5.19–6.18)	0.494
BP (mm Hg)				
Systolic ‡	130 (120–140)	130 (120–140)	130 (120–140)	0.368
Diastolic ‡	80 (80–80)	80 (80–80)	80 (80–80)	0.345
Cardioprotective medications				
Aspirin/clopidogrel	77 (88.5)	39 (86.7)	38 (90.5)	0.578
Beta-blocker	70 (80.5)	38 (84.4)	32 (76.2)	0.332
CCB	25 (28.7)	12 (26.7)	13 (31.0)	0.659
ACEI/ARB	75 (86.2)	41 (91.1)	34 (81.0)	0.170
Other antihypertensive				
drugs	11 (12.6)	4 (8.9)	7 (16.7)	0.275
Diuretics	41 (47.1)	23 (51.1)	18 (42.9)	0.441
Antianginal drugs	40 (46.0)	20 (44.4)	20 (47.6)	0.767
Atorvastatin	87 (100.0)	45 (100.0)	42 (100.0)	
TC (mmol/L)	4.60 ± 0.12	4.61 ± 0.16	4.60 ± 0.18	0.944
TG (mmol/L) †	1.33 (1.20–1.47)	1.49 (1.28–1.72)	1.16 (1.03–1.32)	0.012
HDL-C (mmol/L)	1.36 ± 0.04	1.30 ± 0.05	1.45 ± 0.06	0.053
LDL-C (mmol/L)	2.66 ± 0.10	2.70 ± 0.15	2.61 ± 0.16	0.685

Data are shown as mean ± SEM or as a number (percentage) unless otherwise specified († geometric mean values (95th CI), ‡ median (interquartile range)); BMI: Body mass index; WC: Waist circumference; WHR: Waist-to-hip ratio; CRP: C-reactive protein; BP: Blood pressure; CCB: Calcium channel blocker; ACEI: Angiotensin-converting enzyme inhibitor; ARB: Angiotensin receptor blocker; TC: Total cholesterol; TG: Triglycerides; HDL-C: High-density lipoprotein cholesterol; LDL-C: Low-density lipoprotein cholesterol.

**Table 2 nutrients-13-00903-t002:** Serum PSCK9 and lipid parameters at baseline, after 8 weeks and 13 weeks of supplementation.

Variable	Placebo (*n* = 44)			Dietary Supplement (*n* = 37)			ANOVA (*p*)		
	Baseline	8 Weeks	13 Weeks	Baseline	8 Weeks	13 Weeks	T	S	T × S
PCSK9 (ng/mL)	245.24 ± 19.03	264.91 ± 14.92	264.54 ± 20.46	233.96 ± 20.75	236.41 ± 16.27	237.80 ± 22.31	0.698	0.218	0.807
TC (mmol/L)	4.61 ± 0.16	4.53 ± 0.16	4.69 ± 0.16	4.60 ± 0.18	4.85 ± 0.17	4.84 ± 0.18	0.192	0.479	0.165
TG (mmol/L) †	1.49 (1.28–1.72)	1.64 (1.43–1.87)	1.55 (1.34–1.79)	1.16 (1.03–1.32)	1.35 (1.17–1.56) *	1.20 (1.03–1.41)	0.007	0.012	0.731
HDL-C (mmol/L)	1.30 ± 0.05	1.37 ± 0.05	1.50 ± 0.06 **^,^^#^	1.45 ± 0.06	1.40 ± 0.06	1.47 ± 0.06	0.007	0.416	0.041
LDL-C (mmol/L)	2.70 ± 0.15	2.36 ± 0.14 *	2.53 ± 0.16	2.61 ± 0.16	2.81 ± 0.16	2.73 ± 0.17	0.761	0.341	0.023
ApoA1 (g/L)	1.41 ± 0.04	1.44 ± 0.04	1.42 ± 0.04	1.48 ± 0.04	1.46 ± 0.04	1.45 ± 0.04	0.638	0.389	0.309
ApoB100 (g/L)	1.87 ± 0.07	1.86 ± 0.08	1.97 ± 0.08	1.84 ± 0.08	2.02 ± 0.09 *	1.98 ± 0.09	0.025	0.632	0.084

Values are expressed as mean ± SEM or † geometric mean values (95th confidence interval), as appropriate. Mixed model ANOVA produced *p* values for effects of: T-time, S-supplementation, T × S-time and supplementation interaction effect; PCSK9: Proprotein convertase subtilisin/kexin type 9; TC: Total cholesterol; TG: Triglycerides; HDL-C: High-density lipoprotein cholesterol; LDL-C: Low-density lipoprotein cholesterol; ApoA1: Apolipoprotein A1; ApoB100: Apolipoprotein B100. * *p* < 0.05 versus baseline; ** *p* < 0.01 versus baseline; # *p* < 0.05 versus 8 weeks (*post-hoc* pairwise comparisons with Bonferroni correction).

## Data Availability

The data presented in this study are available on request from the corresponding author. The data are not publicly available due to privacy restrictions.

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
