# Peer review of "Supplementation with Octacosanol Affects the Level of PCSK9 and Restore Its Physiologic Relation with LDL-C in Patients on Chronic Statin Therapy"

_nutrients, 2021, doi:10.3390/nu13030903_

Round 1

Reviewer 1 Report

Dr Ciric and colleagues examined whether a dietary supplement containing octacosanol and vitamin k2-MK7 affected the PCSK9 level in patients on chronic statin therapy. There was a positive correlation between %change in serum PCSK9 levels and that in LDL-C levels in patients with combination of statin and the dietary supplement from baseline to 8 or 13 week. There are several concerns that need to be addressed.

1) Was the dose of statin the same for all patients? The dose and duration of statin contribute to LDL-C and PCSK9 levels, so authors should indicate them in baseline characteristics.

2) In Table2, it is difficult to understand which group ANOVA was used for. Please add the description.

3) How did the authors calculate % Change? Looking at the data, I wonder if the% change of 200 or 500 is correct in PCSK9 levels.

4)In page 7, line 223-225, authors described that the correlation of %change in PCSK9 levels to %change in HDL-C levels (from baseline to endpoint) was not significant in both groups (Supplementary Figure 1). Was there a negative correlation between %change in serum PCSK9 levels and that in HDL-C levels in patients with combination of statin and the dietary supplement from baseline to 8 week? The reviewer thinks that this is one of results in this study.

5) The authors selected patients with favorable changes in lipid profile defined as decrease in LDL-C and increase in HDL-C after 13 weeks. Was there a correlation between LDL-C or HDL-C and PCSK9 level in the selected patients?

6) The text is sometimes difficult to read, so please proofread it in English.

7) Why did the author use a supplement containing octacosanol and vitamin k2 for this study?

Did the author intend that it would be more effective to take Vitamin K2 and Octa together for patients with statin?

8) The authors described that it was found that PCSK9 levels had a trend to be lower at week 13 in the DS group~ in Abstract. Data that did not have significantly difference should not be discussed as the main data. The author should consider increasing the number of subjects.

Reviewer 2 Report

The manuscript is very well written and easy to follow, the study was done taking all the parameters into consideration. Data are clearly presented and properly discussed. The authors have pointed out all the findings and limitations to the research is well appreciable.  

My only concern is the title of the manuscript. The title should be a statement. I don’t find any reason to keep the title with a question mark.

I think that this paper can be accepted for publication on Nutrients. I have no further concerns.

Round 2

Reviewer 1 Report

The manuscript was improved by revision.